# Enhanced voice recognition in musicians

**Allison J. Sletcher, Stefania S. Moro, Jennifer K. E. Steeves***

Department of Psychology and Centre for Vision Research, York University, Toronto, Canada

* steeves@yorku.ca

## Abstract

Musicians typically have extensive auditory experience and demonstrate better pitch, timbre, and tempo discrimination compared to non-musicians. Musical training is also correlated with earlier and more robust cortical and subcortical responses to linguistic stimuli. We asked whether musical expertise may contribute to other auditory tasks, namely person and object recognition when both auditory and visual cues to identity are available. Musicians and non-musicians learned face-voice and car-horn "identity" pairs. Using a forced choice, old/new paradigm, participants were tested for recognition of the learned stimuli presented among distractor stimuli under three stimulus conditions (auditory only, visual only, and bimodal audiovisual). Compared to non-musicians, musicians were more sensitive at recognizing voices but not object sounds. Further, voice recognition sensitivity was positively correlated with both years of musical training and hours of weekly practice suggesting an influence of experience on performance. This differential performance for people and object stimuli is consistent with distinct neural substrates for face and object processing. Overall, this study demonstrates that experience in a sensory domain can benefit aspects of that sensory ability, such as voice but not object sound recognition, likely due to plasticity in distinct neural processing pathways.

## Introduction

Humans can easily recognize people in an instant when seeing a familiar face or hearing a familiar voice. Recognition skills begin even before birth, when a fetus responds preferentially to their mother's voice compared to a stranger's [1]. Shortly after birth, infants respond preferentially to their mother's face [2] and show a preference for faces and voices over other stimuli [3, for review see 4]. Person recognition skills continue to develop through childhood and adolescence until fully mature in adulthood [5,6]. Individual differences in person recognition become more apparent in adulthood when recognition skills reach maturity [7].

Face and voice recognition ability can vary across individuals. On one hand, an individual may be unable to recognize face identity, gender, and facial expression (prosopagnosia) [8–12] or may have the inability to recognize voices (phonagnosia)

**Data availability statement:** All data files are available from the Borealis database: "Dataset of person and object recognition sensitivity in musicians and non-musicians", https://doi.org/10.5683/SP3/L8OBJW, Borealis, V1, UNF:6:nO2uLPSRBi1bGiDk8yYKoQ== [fileUNF]

**Funding:** This research was funded by Canada First Research Excellence Fund: Vision Science to Application (VISTA) (CFREF-2015-00013); Natural Sciences and Engineering Research Council of Canada (NSERC) (327588); Canada Foundation for Innovation (CFI) (12807) to author JKES. The funders had no role in study design, data collection and analysis, decision to publish, or preparation of the manuscript.

**Competing interests:** The authors have declared that no competing interests exist.

[13,14]. On the other hand, some individuals have superior face [15–17] and voice recognition skills [18,19]. Superior voice recognition may be associated with accentuated auditory skills and specialization of auditory cortical pathways through training. For example, highly trained forensic voice experts have superior voice discrimination skills when asked to identify a target voice from a "lineup" of voices [20,21]. Linguists, speech pathologists, and musicians who spend more time evaluating auditory stimuli than the average person may also become auditory specialists [19]. Musicians have established skills in pitch, timbre, and tempo of music [22,23]. Compared to non-musicians, musicians have superior ability to discriminate pitch and fundamental frequencies [24–26], timbre differences [24–27], and tempo [24,25] in both linguistic and musical stimuli.

These cues are common to both linguistic and voice processing and could provide a strong foundation for musicians to develop superior voice recognition. Speech and language tasks, such as speech-in noise recognition (i.e., "cocktail party" scenarios) may be improved by musical training, by strengthening shared resources [for review see 28] through increasing the listening capacity in both ideal acoustic conditions and also difficult acoustic environments [for review see 29]. Training studies on speech-in noise perception [28] and experience dependent plasticity in the auditory system [30–32] suggest that musical training can provide long lasting benefits to auditory function including simple perceptual enhancements and factors impacting higher order cognition such as working memory and intelligence [for review see 33,34]. However, genetic [35] and epigenetic factors (such as behavioural traits like personality [36], motivation [37], and the interaction between factors [38]) are also likely to contribute to musical and speech-in noise recognition [35].

Musical training appears to be associated with plasticity in both cortical and subcortical brain regions that process pitch, duration, and onset time of voice stimuli [for review see 39]. For example, when presented with linguistic pitch patterns musicians have enhanced and more accurate frequency encoding in the inferior colliculus [40]. It has also been shown that musicians have modulated inter-regional neural communication compared to non-musicians [41,42]. For example, in music/ speech categorical processing musicians had increased activation in early primary auditory cortex whereas less experienced non-musicians had increased activation in downstream, higher-order linguistic brain areas such as the inferior frontal gyrus [41]. These findings indicate that cortical and subcortical processes may be under different learning timescales or constraints whether in terms of short-term experiences or for long-term experiences such as musical training. Recent evidence suggests that the enhanced frequency-following response (a scalp-recorded neuroelectric brain recording that serves as a neural index of sound encoding in EEG) may be impacted by myogenic responses, such as a postauricular muscle artifact which may contribute to the musician related differences observed in frequency-following responses [43].

Cortical and subcortical responses have been correlated to years of musical training [39,40,44] establishing a relationship with length of musical training. A relationship between musical training and speech processing at a cortical level has also been demonstrated, where enhanced and earlier cortical responses to syllabic duration

and voice onset time has also been measured in children [45,46]. Finally, years of musical training is positively correlated to faster learning of voice identification in a non-native language [47]. English and Mandarin speaking musicians have an advantage for voice identity recognition for unfamiliar languages that may be attributed to superior pitch processing abilities [48]. While years of music training relationships are suggestive of a training effect, recent studies have demonstrated that inherent, genetic predispositions may also contribute to differences in neural responses [49], speech processing behaviours [50], voice emotion recognition [51], the desire to pursue musical activities [52], and to commit to training for longer than less musically inclined peers [53, for review].

The specialization of auditory processing in musicians may also extend to sound recognition more broadly. Musicians demonstrate better auditory memory for spoken words and environmental sounds [54, for review see 55], but they do not demonstrate better visual memory for object categories [35,37]. Cohen and colleagues (2011) found musicians had better auditory memory for spoken words and environmental sounds compared to non-musicians for an object category recognition task. Typically, in object recognition studies, objects are presented from a range of object categories followed by a memory recognition task to investigate the perceptual processing of general object recognition as opposed to recognition of a specific object representation [54,56]. To date, there have been no studies investigating whether musicians have enhanced auditory sensitivity for specific object identity recognition. The current study asks whether musical expertise contributes to person and object recognition when both auditory and visual cues to identity are available.

## Methods

### Participants

Participants reported normal hearing, normal or corrected-to-normal visual acuity, and no health issues related to vision or hearing. Visual acuity was assessed with an EDTRS eye chart (Precision Vision™, La Salle, IL). All participants identified English as their first language or were early bilingual with English learned before the age of five years. Musicians and non-musicians were recruited through the York University Undergraduate Research Participant Pool (URPP) and the surrounding community. The recruitment period began on July 18, 2022 and ended on February 29, 2024. The study was approved by the Office of Research Ethics at York University and all participants provided informed consent.

**Musicians.** Thirty-five participants [mean age = 21 years (SD = 6 years); mean years of music training = 13 years (SD = 3 years); mean hours of weekly practice = 14 hours/week (SD = 11 hours/week)] self-identified as professional or semi-professional musicians. Musicians reported 8 years or more of musical training beginning before the age of ten years and participated in a regular weekly music practice.

**Non-musicians.** Thirty-five non-musicians participated as control participants. Non-musicians [mean age = 21 years (SD = 8 years); mean years of music training = 0.06 years (SD = 0.04 years)] self-identified as non-musicians, with less than one year of music training, and no current music practice.

### Stimuli

All stimuli have been used previously in similar person and object recognition studies in patients with visual agnosia or unilateral eye enucleation [57,58]. See Hoover et al. [57] and Moro et al. [58] for more detailed stimulus information.

**Person recognition task.** Visual stimuli consisted of 50 greyscale images of female faces, cropped into an oval to eliminate hairlines, with all identifiable markers such as moles and nose rings digitally removed. All face images pictured in this manuscript have given written informed consent (as outlined in PLOS consent form) for use in publication. Auditory stimuli were 50 female voices speaking the same ten second neutral phrase in English. Each auditory clip was controlled for amplitude, background noise, and distinguishing markers such as long pauses and mispronounced words (average sound pressure level (SPL) = 50.6dB (44.2dB − 57.8dB)).

**Object recognition task.** Visual stimuli consisted of 50 greyscale cars presented from the same angle, with all identifiable markers such as ornaments, markings, and license plates removed. Auditory stimuli were 50 unique horn sounds. Each auditory clip was ten seconds in length and controlled for amplitude (average SPL = 53.5 dB (43.9dB − 59.6dB)).

## Procedure

Following the same paradigm as previous studies [57,58] stimuli were presented using Inquisit 6.6.0 [59] on a 23-inch computer display positioned 60 cm from the participant in a dimly lit room and using SONY noise cancelling headphones (Model #: MDR-ZX110NC). Participants responded with a choice of two designated keys on a computer keyboard. Person recognition trials consisted of ten faces paired with ten designated voices to be learned as an "identity" pair. Object recognition trials consisted of ten cars paired with ten designated horns to be learned as an "identity" pair. *Learning phase:* Participants were instructed that they would be tested on their ability to recognize the ten identities, made up of a unique face-voice or car-horn pair and that they had to learn to associate each specific face/car with its corresponding voice/horn, respectively. Identity pairs were presented for a total of four repetitions. Each presentation began with a fixation cross (500 ms) followed by the stimulus pair (10 s) (see Fig 1A and 2A). *Pre-test phase:* To practice their knowledge of recently learned identity pairs and to ensure an adequate level of learning of the paired identities, participants were presented with two learned visual stimuli on the screen and heard one learned auditory stimulus (see Fig 1B and 2B). They were instructed to press the corresponding key on the keyboard for the visual stimulus that was paired to the appropriate auditory stimulus that created an identity pair. Participants could not advance to the next trial until they responded with the correct key. Each identity pair was presented twice with an unrelated learned visual stimulus. *Testing phase:* Following the pre-test recognition was measured in two different blocks: 1. unimodal (visual only and auditory only) and 2. bimodal (face-voice or car-horn pairs). The unimodal block consisted of visual and auditory stimuli presented alone (see Fig 1C and 2C). Stimuli were presented in random order with learned stimuli presented twice and 20 new visual and 20 new auditory distractor stimuli for a total of 80 trials. Participants were asked to press a corresponding key on the keyboard if the stimulus was "learned" or "new". In the bimodal block, participants were presented with combined visual and auditory stimuli. This block consisted of two presentations of each learned identity pair and 30 distractor pairs. Four different stimuli combinations were presented: (1) a learned visual and learned auditory identity pair (congruent learned pair); (2) a new visual stimulus and a new auditory stimulus that are both unfamiliar (congruent new pair); (3) a learned visual stimulus with a new auditory stimulus (incongruent new pair); and (4) a learned auditory stimulus with a new visual stimulus (incongruent new pair) (see Fig 1D and 2D). Participants were asked to press a corresponding key on the keyboard if the stimulus pair was "learned" or "new". Key responses were counterbalanced across participants.

## Results

Participant sensitivity scores were calculated for all conditions (visual only, auditory only, overall bimodal, congruent bimodal, and incongruent bimodal). Overall bimodal sensitivity scores were calculated based on both congruently and incongruently paired stimuli. Participants with perfect scores were adjusted based on recommendations by Macmillan and Creelman [61], by adjusting their score by a constant value (-0.025 in the case where participants scored 100). Approximately 10% of the data for the person identification task and 9% of the data for the object identification task were adjusted in this manner. All statistical analyses were conducted using IBM SPSS Statistics 29 and jamovi 2.3.28.

### Person identity recognition performance

A 2x3 mixed-design analysis of variance (ANOVA) was performed on person identity recognition sensitivity scores with Group (Musicians vs Non-musicians) as the between-subject factor and the Stimulus Condition (Auditory vs Visual vs Bimodal) as the within-subject factor. Data met the assumptions of sphericity (W = 0.97, $p = 0.37$) and homogeneity of

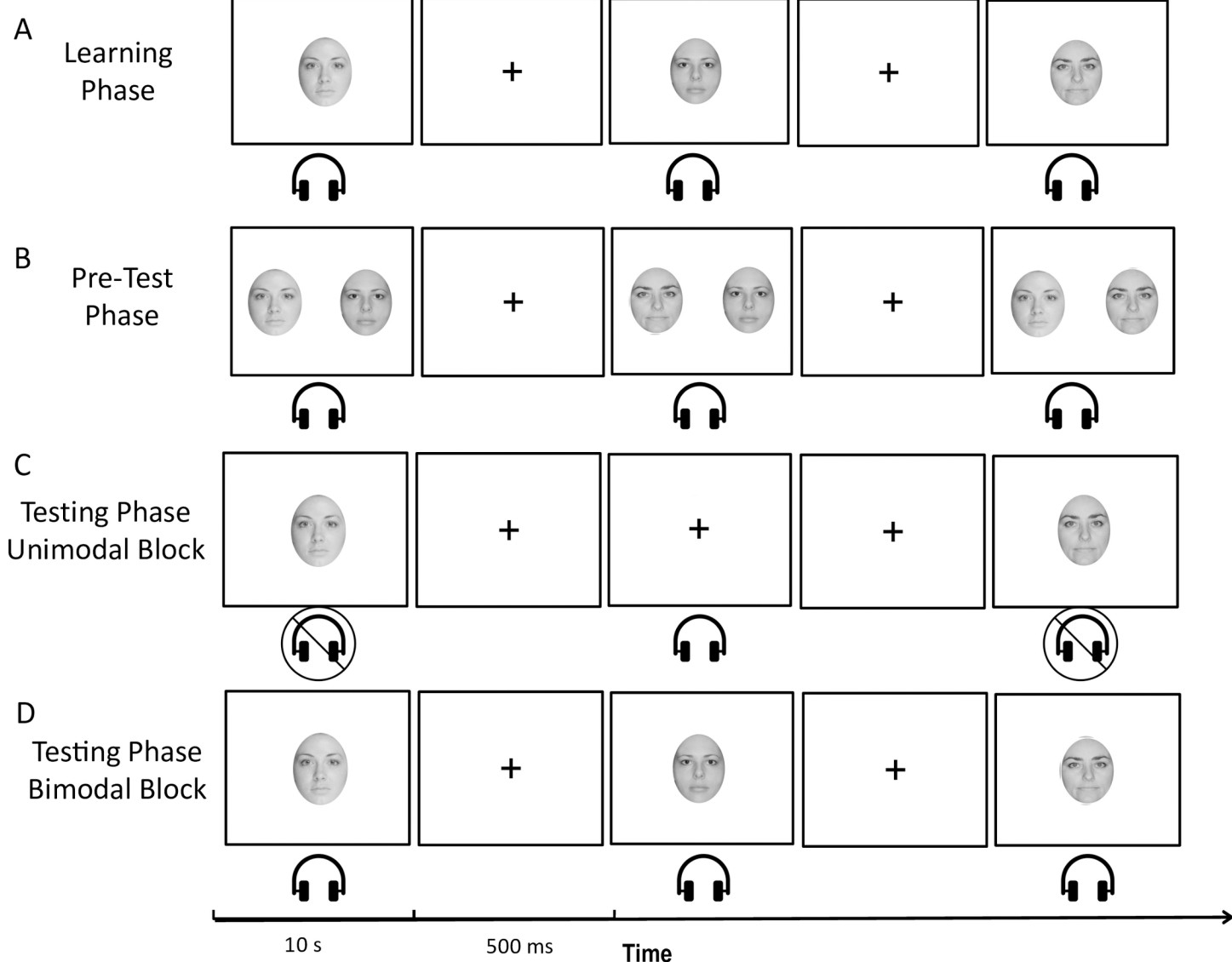

**Fig 1. Schematic diagram representing the person identity recognition procedure adapted from Moro et al. [ 58]** A. Learning phase. B. Pre-test phase. C. Testing phase, unimodal block. D. Testing phase, bimodal block.

variances (Unimodal Visual: $F(1, 68) = 0.11$, $p = 0.74$; Unimodal Auditory: $F(1, 68) = 0.02$, $p = 0.89$; and Bimodal: $F(1,68) = 0.53$, $p = 0.47$) and therefore a parametric hypothesis test was conducted. There was a main effect of Group, $F(1, 68) = 5.40$, $p = 0.02$, $\eta_p^2 = 0.07$, where recognition sensitivity was higher for musicians compared to non-musicians (Fig 3). There was a main effect of Stimulus Condition, $F(2, 136) = 102.88$, $p < 0.001$, $\eta_p^2 = 0.60$. No significant interaction between Group and Stimulus Condition for person identity recognition was found, $F(2, 136) = 0.62$, $p = 0.54$, $\eta_p^2 = 0.009$.

To assess whether musicians are more sensitive to auditory, visual, or bimodal stimuli compared to non-musicians Bonferroni corrected post hoc comparisons were conducted between Groups for each Stimulus Condition. In the auditory condition, musicians ($M = 2.38$, $SD = 0.54$, 95% CI [2.20, 2.57]) had higher sensitivity compared to non-musicians ($M = 2.05$, $SD = 0.61$, 95% CI [1.84, 2.26]), $t(68) = 2.41$, $p = 0.02$, $d = 0.58$. There were no differences between groups in the bimodal

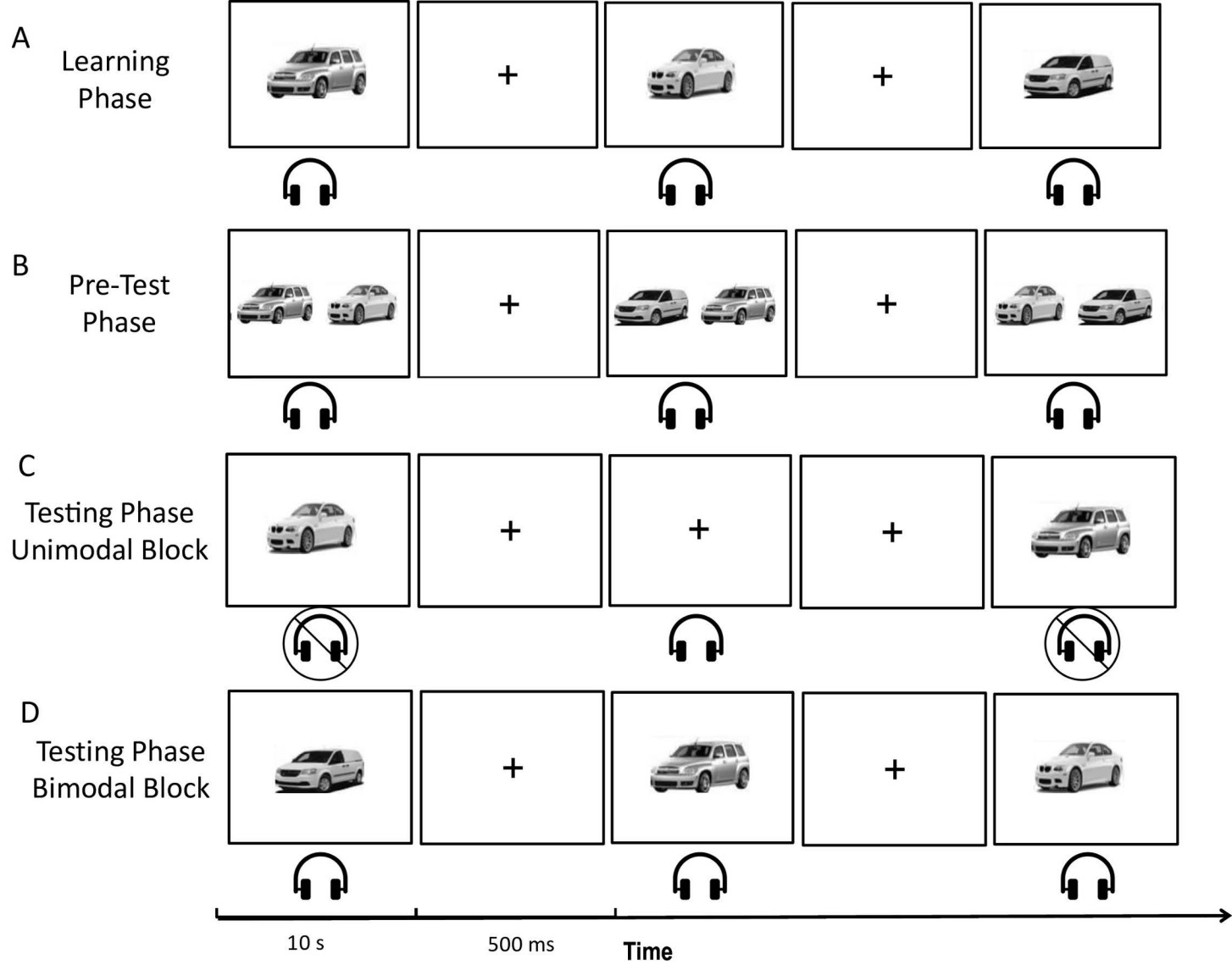

**Fig 2. Schematic diagram representing the object identity recognition procedure adapted from Moro et al. [ 58]** A. Learning phase. B. Pre-test phase. C. Testing phase, unimodal block. D. Testing phase, bimodal block. Car images in this Figure were taken from the open-access Stanford Cars Dataset [60] and are used for illustrative purposes.

condition (musicians: $M = 2.81$, SD = 0.62, 95% CI [2.60, 3.02]; non-musicians: $M = 2.62$, SD = 0.54, 95% CI [2.43, 2.80]), $t(68) = 1.40$, $p = 0.17$, $d = 0.33$, or the visual condition (musicians: $M = 3.44$, SD = 0.52, 95% CI [3.26, 3.62]; non-musicians: $M = 3.27$, SD = 0.57, 95% CI [3.08, 3.47]), $t(68) = 1.30$, $p = 0.20$, $d = 0.31$.

Bonferroni corrected post hoc comparisons were conducted to investigate the difference between Stimulus Conditions. Sensitivity was lower for auditory ($M = 2.21$, SD = 0.60, 95% CI [2.07, 2.36]) compared to visual stimuli ($M = 3.36$, SD = 0.55, 95% CI [3.22, 3.49]), $t(68) = -14.03$, $p < 0.001$, $d = -1.68$ and bimodal stimuli ($M = 2.71$, SD = 0.58, 95% CI [2.58, 2.85]), $t(68) = -5.93$, $p < 0.001$, $d = -0.71$. Visual sensitivity was higher compared to bimodal sensitivity, $t(68) = -8.85$, $p < 0.001$, $d = -1.06$. This indicates that performance was higher overall for the visual domain as participants were more sensitive to visual stimuli compared to auditory stimuli.

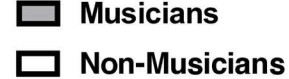

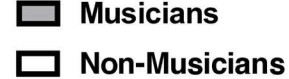

**Fig 3. Person recognition sensitivity scores for each Stimulus Condition (Auditory vs Visual vs Bimodal) according to Group (Musicians vs Non-musicians).** Musicians had better voice recognition sensitivity compared to non-musicians. Error bars represent the standard error of the mean.

### Object identity recognition performance

A 2x3 mixed-design ANOVA was performed on object identity recognition sensitivity scores of between-subject Groups (Musicians vs Non-musicians) and within-subject Stimulus Condition (Visual vs Auditory vs Bimodal) (Fig 4). Data met the assumptions of sphericity (W = 0.986, $p = 0.619$) and homogeneity of variances (Unimodal Visual: $F(1, 68) = 0.006$, $p = 0.939$; Unimodal Auditory: $F(1, 68) = 0.003$, $p = 0.957$; and Bimodal: $F(1,68) = 4.427$, $p = 0.04$) and therefore a parametric hypothesis test was conducted. Unlike for person recognition, there were no Group differences for object recognition, $F(1, 68) = 0.61$, $p = 0.44$, $\eta_p^2 = 0.009$. Similar to person recognition, there was a main effect of Stimulus Condition,

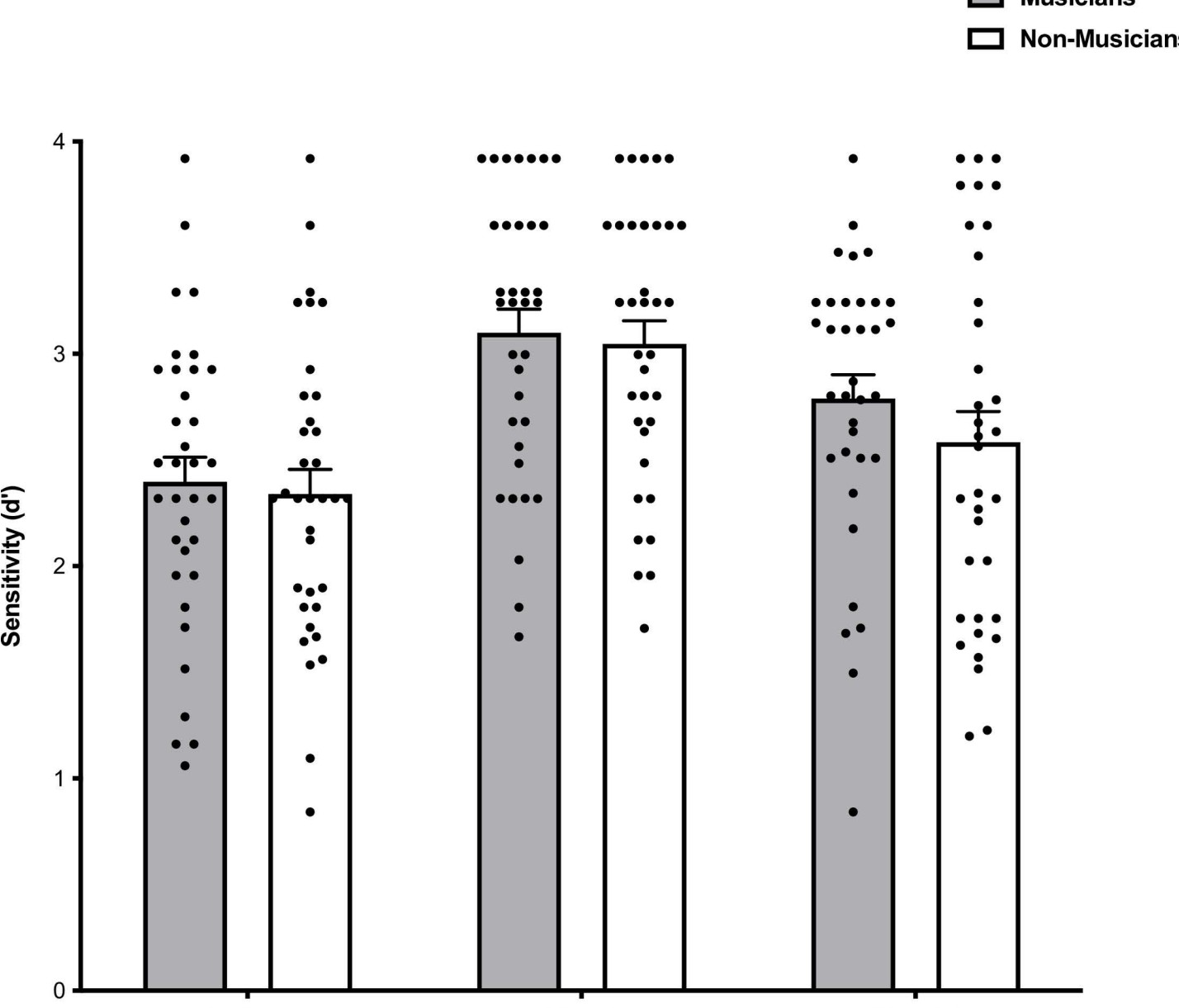

**Fig 4. Overall object recognition sensitivity scores for each Stimulus Condition (Auditory vs Visual vs Bimodal) according to Group (Musicians vs Non-musicians).** No differences were found between groups in object recognition sensitivity across all modalities. Error bars represent standard error of the mean.

$F(2, 136) = 34.52$, $p < 0.001$, $\eta_p^2 = 0.34$, and no interaction between Group and Stimulus Condition, $F(2, 136) = 0.53$, $p = 0.59$, $\eta_p^2 = 0.008$.

Bonferroni corrected post hoc pairwise comparisons of Stimulus Condition indicated that sensitivity for the visual stimuli ($M = 3.07$, SD = 0.65, 95% CI [2.92, 3.23]), was higher compared to the auditory stimuli ($M = 2.37$, SD = 0.68, 95% CI [2.21, 2.53]), $t(68) = 8.41$, $p < 0.001$, $d = 1.01$, and the bimodal stimuli ($M = 2.69$, SD = 0.77, 95% CI [2.50, 2.87]), $t(68) = 4.77$,

$p < 0.001$, $d = 0.57$. Sensitivity for the bimodal stimuli was higher compared to the auditory stimuli, $t(68) = 3.56$, $p = 0.002$, $d = 0.42$. Similar to person recognition, performance was higher overall for the visual domain compared to the auditory and bimodal domains for object recognition across groups.

### Relationship between musical experience and auditory sensitivity

Spearman rank-order correlation analyses were conducted to examine the relationship between musical experience and auditory sensitivity for musicians and non-musicians. The musical experience variable (Years of Musical Training) was not normally distributed therefore a non-parametric analysis was conducted (Non-musician: $W = 0.25$, $p < 0.001$; Musicians: $W = 0.94$, $p = 0.07$). There was a small positive correlation between years of training and performance sensitivity, where more training experience was related to higher sensitivity for voices in person recognition. Further, there was no relationship between years of training and object sound recognition across all stimulus condition modalities. See Table 1 for a breakdown of the correlation results.

### Relationship between hours of weekly practice and auditory sensitivity

Spearman rank-order correlation analyses were conducted to examine the relationship between hours of weekly musical practice and auditory sensitivity for musicians and non-musicians. The hours of weekly musical practice variable was not normally distributed therefore a non-parametric correlation analysis was conducted (Non-musician: $W < 0.001$, $p < 0.001$; Musicians: $W = 0.830$, $p < 0.001$). There was a moderate positive correlation between hours of weekly practice and performance sensitivity, where increased hours of weekly practice was related to higher sensitivity for voices in person recognition. Further, there was no relationship between hours of weekly practice and object sound recognition across all stimulus condition modalities. See Table 1 for detailed Spearman correlation results.

### Comparison across voices and horns

To determine whether there was an effect of the type of auditory stimulus (Voices or Horns) we conducted parametric paired samples t-tests to compare the auditory sensitivity scores of Musicians and Non-musicians across the person identity and object identity tasks (Fig 5). Musicians did not differ in their auditory sensitivity for voices ($M = 2.38$, $SD = 0.54$, 95% CI [2.20, 2.57]) compared to object sounds ($M = 2.40$, $SD = 0.66$, 95% CI [2.16, 2.63]), $t(34) = -0.125$, $p = 0.90$, $d = -0.02$. Non-musicians demonstrated a strong trend for enhanced auditory sensitivity for object sounds ($M = 2.34$, $SD = 0.69$, 95% CI [2.10, 2.58]) compared to voices ($M = 2.05$, $SD = 0.61$, 95% CI [1.84, 2.26]), $t(34) = -2.031$, $p = 0.05$, $d = -0.34$.

**Table 1. Spearman rank-order correlations examining the relationship between years of training and hours of weekly practice for auditory person and object identity recognition sensitivity.** Significant correlations are indicated with an asterisk ($p$-values are not adjusted for multiple comparisons).

| | Auditory Sensitivity | |
|---|---|---|
| | $\rho$ | $p$ |
| | **Years of Training** | |
| **Person Identity Recognition** | 0.26 | 0.03* |
| **Object Identity Recognition** | 0.05 | 0.67 |
| | Hours of Weekly Practice | |
| **Person Identity Recognition** | 0.30 | 0.01* |
| **Object Identity Recognition** | 0.07 | 0.57 |

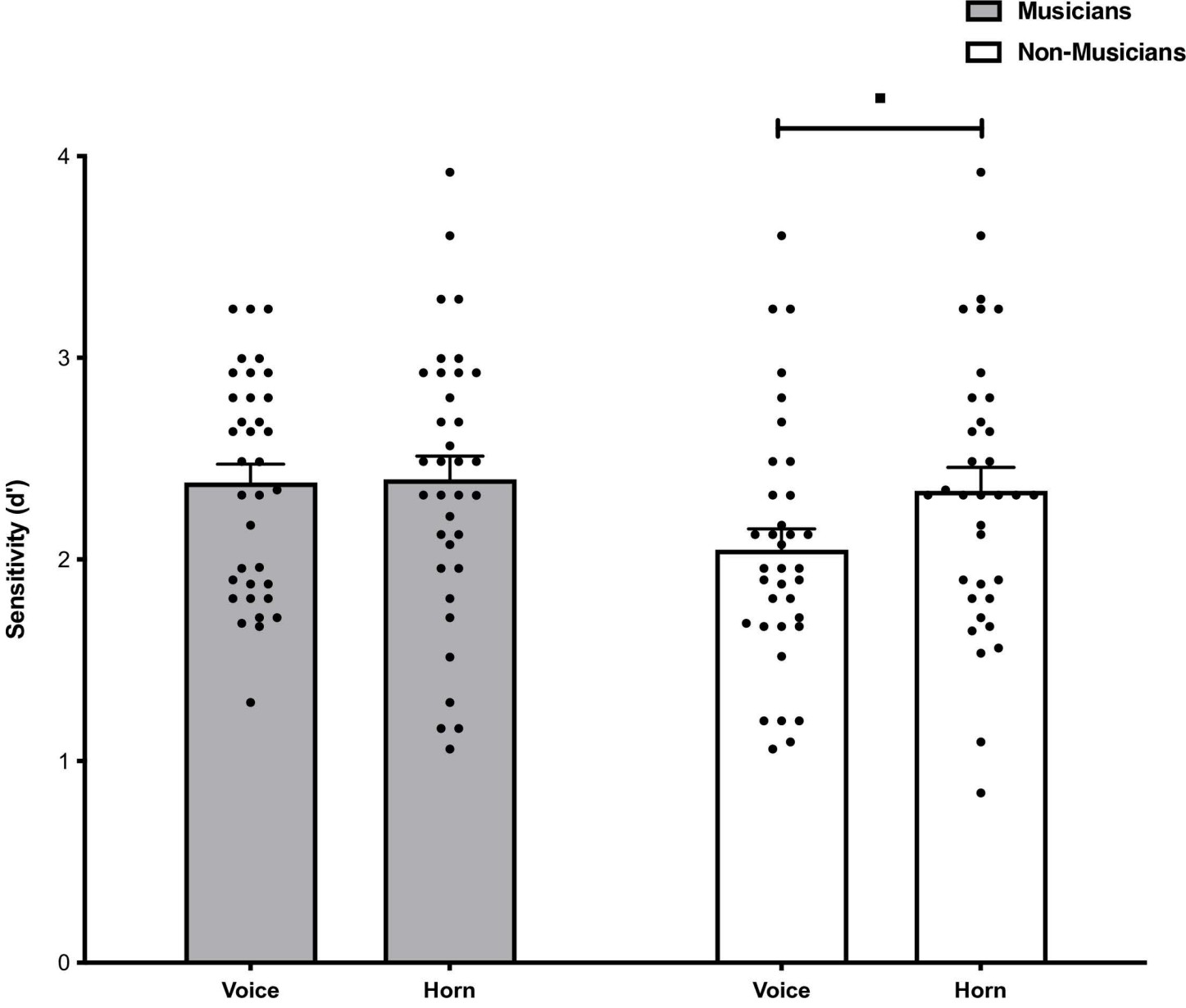

**Fig 5. Auditory only stimulus sensitivity in the person (face/voice) and object (car/horn) identity tasks for Musicians (grey bars) and Non-musicians (white bars).** Error bars represent standard error of the mean.

## Discussion

We demonstrated that musicians from our sample have better sensitivity for recognition of voices compared to non-musicians from our sample and that this ability is related to the extent of experience as measured by hours of weekly practice and years of training. These findings suggest that learned skill in auditory processing through musical specialization contributes to driving group differences between musicians and non-musicians. Further, our study demonstrated that the enhancement of auditory sensitivity for voice perception is not replicated in auditory sensitivity for object sound

perception. This indicates that person identity processing is a unique process and that the auditory advantage of musical specialization may not broadly translate to other domains such as object recognition, specifically for car-horn identification. Finally, our study demonstrated that better unimodal auditory sensitivity for voices is positively correlated with years of musical experience, as well as hours of practice indicating that learned skill and expertise impact voice recognition for musicians.

These findings suggest that musicians have specialized auditory skills that transfer to voice recognition and are aligned with previous studies demonstrating that musicians have enhanced recognition of timbre, pitch, and fundamental frequencies [24–26,62], which may be supported by enhanced cortical and subcortical processing of auditory information [39,40,44]. On one hand, musicians enhanced auditory sensitivity was restricted to voices in the person recognition task and did not translate to enhanced auditory sensitivity in the object identity recognition task. On the other hand, non-musicians show a strong trend for better sensitivity to object sounds compared to voices, which aligns with previously reported data [57]. These results indicate that musical expertise may only benefit certain aspects of sound processing. This is consistent with the notion of distinct neural substrates for object and person processing [11,57,63–69]. Our findings are consistent with prior studies of patients demonstrating that person compared to object identity recognition can be differentially affected by clinical vision disorders [57,58,70]. Clinical case studies have shown that lesions to discrete regions within visual cortex can lead to specific deficits in face recognition (prosopagnosia) [10–12,57,70,71], and in object recognition (object agnosia) [72–74]. Similarly in the auditory domain, clinical case studies have shown lesions in auditory cortex can result in impaired recognition of melodic tunes or tone deafness (amusia) and voice recognition (phonagnosia) while sparing environmental sounds recognition [75]. Our data are consistent with such clinical cases [75,76].

Voice recognition sensitivity was positively correlated with both years of training and hours of weekly practice indicating an influence of experience through a learned skill. Both behaviour and brain response differences in musicians is related to years of musical training. For example, faster voice processing in unfamiliar languages and enhanced cortical and subcortical responses to auditory stimuli has also been correlated to years of musical training in musicians [39,40,44,47]. These relationships indicate that the development of auditory expertise through experience can influence neuroplastic changes in the brain. Future research should evaluate the neural correlates of enhanced auditory sensitivity for voices compared to object sounds in musicians.

In conclusion, compared to non-musicians, musicians have enhanced sensitivity for voice identity recognition but not object sound recognition. This auditory specialization is positively correlated with both years of musical training and hours of weekly practice indicating an influence of experience through a learned skill. Overall, this study demonstrates the contribution of musical experience towards enhancement of a specific sensory ability, such as voice but not object sound recognition likely due to plasticity in distinct neural processing pathways.

## Acknowledgments

We sincerely thank all participants for taking part in this study.

## Author contributions

**Conceptualization:** Allison J. Sletcher, Stefania S. Moro, Jennifer K. E. Steeves.

**Data curation:** Allison J. Sletcher.

**Formal analysis:** Allison J. Sletcher.

**Funding acquisition:** Jennifer K. E. Steeves.

**Investigation:** Allison J. Sletcher.

**Methodology:** Stefania S. Moro, Jennifer K. E. Steeves.

**Project administration:** Stefania S. Moro, Jennifer K. E. Steeves.

**Resources:** Jennifer K. E. Steeves.

**Supervision:** Stefania S. Moro, Jennifer K. E. Steeves.

**Validation:** Stefania S. Moro, Jennifer K. E. Steeves.

**Visualization:** Stefania S. Moro.

**Writing – original draft:** Allison J. Sletcher.

**Writing – review & editing:** Stefania S. Moro, Jennifer K. E. Steeves.

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
