## [Decision Letter · Decision Letter 0]

22 Oct 2024

PONE-D-24-32230Enhanced voice recognition in musiciansPLOS ONE

Dear Dr. Moro,

Thank you for submitting your manuscript to PLOS ONE. After careful consideration, we feel that it has merit but does not fully meet PLOS ONE’s publication criteria as it currently stands. A major revision is requested. Therefore, we invite you to submit a revised version of the manuscript that addresses the points raised during the review process.

We look forward to receiving your revised manuscript.

Kind regards,

Gavin Bidelman, Ph.D.

Academic Editor

PLOS ONE

Journal Requirements:

1. When submitting your revision, we need you to address these additional requirements. Please ensure that your manuscript meets PLOS ONE's style requirements, including those for file naming. The PLOS ONE style templates can be found at https://journals.plos.org/plosone/s/file?id=wjVg/PLOSOne_formatting_sample_main_body.pdf and https://journals.plos.org/plosone/s/file?id=ba62/PLOSOne_formatting_sample_title_authors_affiliations.pdf 2. Thank you for stating the following financial disclosure: "Canada First Research Excellence Fund: Vision Science to Application (VISTA) (CFREF-2015-00013); Natural Sciences and Engineering Research Council of Canada (NSERC) (327588); Canada Foundation for Innovation (CFI) (12807). " Please state what role the funders took in the study.  If the funders had no role, please state: "The funders had no role in study design, data collection and analysis, decision to publish, or preparation of the manuscript." If this statement is not correct you must amend it as needed. Please include this amended Role of Funder statement in your cover letter; we will change the online submission form on your behalf. 3. Thank you for stating the following in the Acknowledgments Section of your manuscript: "We sincerely thank all participants for taking part in this study. This research was supported by grants from Canada First Research Excellence Fund: Vision Science to Application (VISTA) (CFREF-2015-00013); Natural Sciences and Engineering Research Council of Canada (NSERC) (327588); Canada Foundation for Innovation (CFI) (12807) to JKES." We note that you have provided funding information that is not currently declared in your Funding Statement. However, funding information should not appear in the Acknowledgments section or other areas of your manuscript. We will only publish funding information present in the Funding Statement section of the online submission form.  Please remove any funding-related text from the manuscript and let us know how you would like to update your Funding Statement. Currently, your Funding Statement reads as follows: "Canada First Research Excellence Fund: Vision Science to Application (VISTA) (CFREF-2015-00013); Natural Sciences and Engineering Research Council of Canada (NSERC) (327588); Canada Foundation for Innovation (CFI) (12807)." Please include your amended statements within your cover letter; we will change the online submission form on your behalf. 4. We note that Figure 1 includes an image of a participant in the study. As per the PLOS ONE policy (http://journals.plos.org/plosone/s/submission-guidelines#loc-human-subjects-research) on papers that include identifying, or potentially identifying, information, the individual(s) or parent(s)/guardian(s) must be informed of the terms of the PLOS open-access (CC-BY) license and provide specific permission for publication of these details under the terms of this license. Please download the Consent Form for Publication in a PLOS Journal (http://journals.plos.org/plosone/s/file?id=8ce6/plos-consent-form-english.pdf). The signed consent form should not be submitted with the manuscript, but should be securely filed in the individual's case notes. Please amend the methods section and ethics statement of the manuscript to explicitly state that the patient/participant has provided consent for publication: “The individual in this manuscript has given written informed consent (as outlined in PLOS consent form) to publish these case details”.  If you are unable to obtain consent from the subject of the photograph, you will need to remove the figure and any other textual identifying information or case descriptions for this individual.

Reviewers' comments:

Reviewer's Responses to Questions

**Comments to the Author**

1. Is the manuscript technically sound, and do the data support the conclusions?

Reviewer #1: Partly

Reviewer #2: Partly

2. Has the statistical analysis been performed appropriately and rigorously? 

Reviewer #1: No

Reviewer #2: Yes

3. Have the authors made all data underlying the findings in their manuscript fully available?

Reviewer #1: Yes

Reviewer #2: Yes

4. Is the manuscript presented in an intelligible fashion and written in standard English?

Reviewer #1: Yes

Reviewer #2: Yes

5. Review Comments to the Author

Reviewer #1: Overview

The goal of this paper is to evaluate whether the neuroplastic benefits of music training on auditory skills extend to object-sound and face-voice recognition following brief identification training. In general, I think this study has the potential to provide nuance to the effects of music training on perceptual abilities. However, I have significant concerns about the statistical analyses that potentially call into question the validity of the results and may limit overall conclusions.

Major comments

Introduction

1. It seems that “voice processing” is used interchangeably with notions of speech processing in the introduction and discussion, but these ideas are quite distinct. While the included citations are used to support theories of musical training effects on the brain, it creates confusion about the motivation for the study and in some cases appears to misinterpret findings across the literature. For example:

a. (lines 63-66) “A relationship between musical training and voice processing at a cortical level has also been demonstrated, where enhanced and earlier cortical responses to syllabic duration and voice onset time has also been measured in children.” But syllabic duration and voice onset refer to processing of speech phonemes, not properties of voices, per se

b. (lines 261-263) “For example, faster learning of voices in unfamiliar languages and enhanced cortical and subcortical responses to auditory stimuli has also been correlated to years of musical training in musicians.” – but several of these citations are about processing speech phonemes, not voice recognition

To aid clarity, I suggest reframing those citations as "speech processing" and include more relevant literature for "voice processing" as it pertains to voice recognition and learning to distinguish timbre or other voice characteristics which would be more pertinent for the current study.

2. Additionally, the introduction does not go far enough in motivating the current research question, in my opinion. One area that would benefit from more details is literature on human voice processing vs. non-human auditory objects. The only sentence I can find in the intro is “Musicians demonstrate better auditory memory for spoken words and environmental sounds,35, for review see 36 but they do not demonstrate better visual memory for object categories.35, 37” (lines 71-73). The introduction up to this point focused mainly on voice recognition (though see previous comment), so it comes a bit as a surprise to me when object recognition is suddenly mentioned in the last paragraph. Additionally, the enhanced skills in “pitch, timbre, and tempo of music” due to music training might also theoretically extend to object recognition. Even though the findings reportedly suggest otherwise, more details in the intro would plant a seed for possible similarities, differences, and musical training effects on processing voices vs. non-voice stimuli.

3. The citations for neural processes & musicianship do not fully reflect the current state of the literature. I recognize the current study is not a neuroimaging study, and the generic statements about neural processes and music training-induced neuroplasticity are merely to support a case for possible transfer effects to voice & object recognition. However, I think the current manuscript is missing (or perhaps misrepresenting) important considerations/nuance from the body of work that was already briefly cited.

a. For example, the intro cites some studies in relation to subcortical processes (mainly FFR; lines 58-62), but more recent work has pointed out the cortical contributions with low stimulus F0s (e.g., Bidelman 2018, NeuroImage)—relevant for the cited older Kraus work—or MEG methodology (e.g., Coffey et al 2016, Nature Communications). Cortical vs. subcortical processes may be under different learning timescales or constraints, whether in terms of the short-term experiences like the experimental design presented here or for longer-term experiences like musical training.

b. Additionally, some of the musicianship-FFR literature may need to be revisited in light of recent evidence that suggests myogenic responses (postauricular muscle artifact) might contribute to the musician-related differences observed in FFRs (Bidelman et al 2024, Frontiers in Neuroscience).

c. (lines 62-63) While years of music training relationships are more suggestive of a training effect rather than mere categorical group differences, it does not rule out inherent, genetic predispositions that could have contributed to differences in neural responses (e.g., Mankel & Bidelman, 2018, PNAS), speech processing behaviors (e.g., Swaminathan & Schellenberg 2017, Psychonomic Bulletin & Review), voice emotion recognition (Correia et al 2022 Emotion), the desire to pursue musical activities in the first place (e.g., Wesseldijk et al 2021, Psychological Science), and/or to stick with the training for longer than less-musically-inclined peers (see also review by Schellenberg & Lima 2024, Annual Review of Psychology).

i. For further discussions on genetic contributions, see: Wesseldijk et al 2023, Translational Psychiatry; Tan et al 2014, Frontiers in Psychology; Mosing et al 2014, Psychological Science; Hambrick & Tucker-Drob 2015, Psychonomic Bulletin & Review; Wesseldijk et al 2019, Developmental Psychology; etc.

Methods

4. Certain aspects of the methods are missing or not well described, making it difficult to evaluate the rigor and potentially limiting reproducibility of the study:

a. Even if the stimuli were controlled for RMS amplitudes, background noise, distinguishing features, etc. as mentioned in the methods, characteristic differences across the stimuli could still artificially inflate identification performance. Were the objects, faces, and associated audio validated for equivalent identification performance? Did all the voices speak the same English phrase? Do the car horns all have unique timbres and pitch contours, or are the pitch contours similar?

i. The cited Hoover et al (2010) previously reported sensitivity differences across the voice and horn stimuli. Thus, it is possible that the stimuli are not equated in identification difficulty across conditions which could impact the group effects. For example, if there is already increased sensitivity to the horn sounds compared to voice recognition, perhaps there was not enough “room” or variation for music training to demonstrate a group-wise effect in this study, even if underlying perceptual advantages were present in this group.

ii. However, it does not seem the same effects are replicated here, at least not to the same degree. It would be pertinent to address these apparent deviations from previous studies using the same stimuli in the discussion.

b. In the bimodal block, it is not clear whether the correct response for an incongruent pair is “new” or if at least one cue is learned then the correct response is “learned”. Based on the previous work by Hoover et al, I think the latter qualifies as a “new” pair, but it would still be helpful if this was more clearly described in the current paper.

c. The stats are missing critical details on data cleaning procedures, ANOVA assumptions (normality, homogeneity of variance), how influential outliers evaluated, etc. Failure to conduct a thorough evaluation of the data and rule out potential statistical assumption violations could contribute to the seemingly discrepant non-significant interaction but significant group x auditory post hoc comparison for the person identity recognition block. A more detailed description of the data checking procedures would at least increase confidence in the results.

d. Based on my understanding of d’, the mean sensitivity scores >2 or >3 are good indicators or near-optimal/ceiling performance across the board. However, line 155 states “Participants with perfect scores were adjusted”. Although a citation is listed at the end of this sentence, I think this is a case where a full description of the exact methodology is needed for readers to understand exactly how scores were adjusted to know its potential impact on the results. How many scores needed adjustment for each condition or in each group?

Results & discussion

5. It is debated whether post hoc tests should be conducted on an interaction term when the global group x condition interaction itself is not significant for person identity recognition. Since the global F-test and the post hoc t-tests technically assess different hypotheses, of course it is theoretically possible to get a significant post hoc without a significant F-test (Midway et al 2020, PeerJ). Yet, others argue that continuing to test post hocs on a non-significant interaction term can further inflate Type I error even if using post hoc correction factors (Nieuwenhuis et al 2011, Nature Neuroscience; Garofalo et al 2022, PLOS ONE). Further, the means of each group are within 1 SD of each other. I am thus highly skeptical of the Group effects in the Auditory-only condition for the voice identity recognition block. Conclusions drawn from this result seem tenuous at best and should therefore be interpreted more carefully.

6. One suggested approach might be to include 95% confidence intervals in addition to (or in place of) the post hoc analyses to reduce likelihood of Type I error (Garofalo et al 2022, PLOS ONE)

7. Running several similar correlations on subsets of the data as in Table 1 without corrections for family-wise multiple comparisons are also at risk for Type I error (Curtin & Schulz 1998, Biological Psychiatry). From what I can tell, a simple Bonferroni correction would make most if not all of the starred correlations in Table 1 no longer significant.

8. In light of the stats/results concerns listed above, I feel the effects of music training on voice/object recognition are not robust enough, or perhaps the experimental paradigm was not quite sensitive enough to these group differences in a manner that survives more stringent multiple comparisons. As a result, this study reads as rather exploratory, and I believe more caution should be expressed throughout the entire manuscript when interpreting the results.

Minor comments

9. (lines 50-53) “For example, linguists, speech pathologists, and musicians who spend more time evaluating auditory stimuli than the average person may become auditory specialists. Highly trained forensic voice experts have superior voice discrimination skills when asked to identify a target voice from a “lineup” of voices. Musicians have established skills…” I think the inclusion of the forensic voice experts sentence creates an awkward flow in this paragraph. The previous sentence is a setup about musicians with additional details about musicians in the sentence afterwards, but the continuity of thought is broken up by the forensic experts sentence. Suggest rephrasing.

10. Is there a model number for the SONY headphones? It is possible that some consumer-grade headphone response properties filter out certain frequencies, perhaps with mildly degraded auditory quality that would otherwise aid discrimination of the auditory stiuli. It would be helpful for future replications of this work to be certain of the auditory equipment used

11. The mean stimulus intensity level for the auditory sounds (or range) should be added to the methods.

12. “visual processing is superior” (line 178), “visual sensitivity is superior” (line 198) is odd phrasing, stating superiority of a modality. I recommend modifying these phrases to something like “performance/accuracy was higher overall for visual domain” or “higher sensitivity to the visual domain” or something similar to avoid the implication of modality “superiority”.

13. I think some of the concerns above about the data would be partly resolved if another plot type was used to show the spread of the data (e.g., a strip chart overlaid on the bar chart, violin plots, histograms next to the bar charts, etc.)

Reviewer #2: Review PONE-D-24-32230

To Editor and Authors

This manuscript describes a behavioural experiment comparing Musician to Non-musician participants at an old/new recognition task with Person (face/voice) or Car (car/honk) stimuli presented in 3 versions: auditory, visual, or both modalities.

Participants first learned an auditory-visual association task with both the Person or Car stimuli, then were tested with the 3 versions (auditory, visual, audiovisual) of the old/new task.

Results indicate a sensitivity difference between musicians and non-musicians for recall of the auditory stimuli only, correlating mildly with years of musical training in Musicians. This result corroborates past results showing that Musicians have better timbre recognition in general.

The article is well-written, with proper reference to the literature, and clean statistical analyses.

A major problem I see is that there is no clear relation to multimodality – the main purpose of the study. Indeed, the audiovisual task could be solved by participants either by detecting deviance from learned associations (true audio-visual association learning) OR by detecting novel stimuli in either modality without regard to cross-modal congruence.

Indeed the task could be solved either by recognizing that the presented association was not one of the learned ones (true bimodal learning) OR by detecting novelty in either modality – a much easier task not relying on audio-visual integration. It is not clear form the data which of these two processes the participants engaged in. This complexity could have been avoided by using incongruent audiovisual pairings composed of old, although incongruent stimuli – not novel ones.

This does not change much about the interesting result of superior voice recognition in musicians, but only seems as a side result given the initial multimodal scope of the project.

6. PLOS authors have the option to publish the peer review history of their article (what does this mean? ). If published, this will include your full peer review and any attached files.

**Do you want your identity to be public for this peer review?** For information about this choice, including consent withdrawal, please see our Privacy Policy .

Reviewer #1: No

Reviewer #2: **Yes: ** Pascal Belin

---

## [Author Response · Author response to Decision Letter 1]

16 Jan 2025

Please see attached response to reviewers document.

Journal Requirements:

All requirements have been addressed.

2. Thank you for stating the following financial disclosure: "Canada First Research Excellence Fund: Vision Science to Application (VISTA) (CFREF-2015-00013); Natural Sciences and Engineering Research Council of Canada (NSERC) (327588); Canada Foundation for Innovation (CFI) (12807). "

The statement is correct and has been added to the cover letter to be updated in the online submission system.

3. Thank you for stating the following in the Acknowledgments Section of your manuscript: "We sincerely thank all participants for taking part in this study. This research was supported by grants from Canada First Research Excellence Fund: Vision Science to Application (VISTA) (CFREF-2015-00013); Natural Sciences and Engineering Research Council of Canada (NSERC) (327588); Canada Foundation for Innovation (CFI) (12807) to JKES."

Please remove any funding-related text from the manuscript and let us know how you would like to update your Funding Statement. Currently, your Funding Statement reads as follows: "Canada First Research Excellence Fund: Vision Science to Application (VISTA) (CFREF-2015-00013); Natural Sciences and Engineering Research Council of Canada (NSERC) (327588); Canada Foundation for Innovation (CFI) (12807)."

Funding information has been removed from our manuscript acknowledgment section. The funding information provided in the funding statement in the online submission form is accurate.

4. We note that Figure 1 includes an image of a participant in the study.

The face image in figure 1 is not a participant. This is an image of a stimulus that was used during the study. All faces used as stimuli have been modified from their original form. Models gave written informed consent to use their images in our study and for subsequent use in publications and presentations. We have included the relevant text above in our manuscript both in text and in the caption for figure 1.

Reviewer #1: Overview

The goal of this paper is to evaluate whether the neuroplastic benefits of music training on auditory skills extend to object-sound and face-voice recognition following brief identification training. In general, I think this study has the potential to provide nuance to the effects of music training on perceptual abilities. However, I have significant concerns about the statistical analyses that potentially call into question the validity of the results and may limit overall conclusions.

Major comments

Introduction

1. It seems that “voice processing” is used interchangeably with notions of speech processing in the introduction and discussion, but these ideas are quite distinct. While the included citations are used to support theories of musical training effects on the brain, it creates confusion about the motivation for the study and in some cases appears to misinterpret findings across the literature. For example:

a. (lines 63-66) “A relationship between musical training and voice processing at a cortical level has also been demonstrated, where enhanced and earlier cortical responses to syllabic duration and voice onset time has also been measured in children.” But syllabic duration and voice onset refer to processing of speech phonemes, not properties of voices, per se

b. (lines 261-263) “For example, faster learning of voices in unfamiliar languages and enhanced cortical and subcortical responses to auditory stimuli has also been correlated to years of musical training in musicians.” – but several of these citations are about processing speech phonemes, not voice recognition

To aid clarity, I suggest reframing those citations as "speech processing" and include more relevant literature for "voice processing" as it pertains to voice recognition and learning to distinguish timbre or other voice characteristics which would be more pertinent for the current study.

Thank you for your suggestions. We have reframed the citations mentioned in points A and B as “speech processing” and have expanded our discussion of voice processing in our introduction. All modifications to the introduction and discussion have been made using track changes. A copy of the added discussion on voice processing has also been copied below:

These cues are common to both linguistic and voice processing and could provide a strong foundation for musicians to develop superior voice recognition. Speech and language tasks, such as speech-in noise recognition (ie. “cocktail party” scenarios) may be improved by musical training, by strengthening shared resourcesfor review see 28 through increasing the listening capacity in both ideal acoustic conditions and also difficult acoustic environments.for review see 29 Training studies on speech-in noise perception28 and experience dependent plasticity in the auditory system30 - 32 suggest that musical training can provide long lasting benefits to auditory function including simple perceptual enhancements and factors impacting higher order cognition such as working memory and intelligence.for review see 33, 34 However, genetic35 and epigenetic factors (such as behavioural traits like personality,36 motivation,37 and the interaction between factors38 are also likely to contribute to musical and speech-in noise cognition.35

2. Additionally, the introduction does not go far enough in motivating the current research question, in my opinion. One area that would benefit from more details is literature on human voice processing vs. non-human auditory objects. The only sentence I can find in the intro is “Musicians demonstrate better auditory memory for spoken words and environmental sounds,35, for review see 36 but they do not demonstrate better visual memory for object categories.35, 37” (lines 71-73). The introduction up to this point focused mainly on voice recognition (though see previous comment), so it comes a bit as a surprise to me when object recognition is suddenly mentioned in the last paragraph. Additionally, the enhanced skills in “pitch, timbre, and tempo of music” due to music training might also theoretically extend to object recognition. Even though the findings reportedly suggest otherwise, more details in the intro would plant a seed for possible similarities, differences, and musical training effects on processing voices vs. non-voice stimuli.

Thank you for your insightful comment. We have expanded the introduction to include a discussion of human voice processing compared to non-human auditory objects. All modifications have been made using track changes. A copy of the added discussion on voice vs object sound processing has also been copied below:

The specialization of auditory processing in musicians may also extend to sound recognition more broadly. Musicians demonstrate better auditory memory for spoken words and environmental sounds,54, for review see 56 but they do not demonstrate better visual memory for object categories.35, 37 Cohen and colleagues (2011) found musicians had better auditory memory for spoken words and environmental sounds compared to non-musicians for an object category recognition task. Typically, in object recognition studies, objects are presented from a range of object categories followed by a memory recognition task to investigate the perceptual processing of general object recognition as opposed to recognition of a specific object representation.54, 56 To date, there have been no studies investigating whether musicians have enhanced auditory sensitivity for specific object identity recognition. The current study asks whether musical expertise contributes to person and object recognition when both auditory and visual cues to identity are available.

3. The citations for neural processes & musicianship do not fully reflect the current state of the literature. I recognize the current study is not a neuroimaging study, and the generic statements about neural processes and music training-induced neuroplasticity are merely to support a case for possible transfer effects to voice & object recognition. However, I think the current manuscript is missing (or perhaps misrepresenting) important considerations/nuance from the body of work that was already briefly cited.

a. For example, the intro cites some studies in relation to subcortical processes (mainly FFR; lines 58-62), but more recent work has pointed out the cortical contributions with low stimulus F0s (e.g., Bidelman 2018, NeuroImage)—relevant for the cited older Kraus work—or MEG methodology (e.g., Coffey et al 2016, Nature Communications). Cortical vs. subcortical processes may be under different learning timescales or constraints, whether in terms of the short-term experiences like the experimental design presented here or for longer-term experiences like musical training.

b. Additionally, some of the musicianship-FFR literature may need to be revisited in light of recent evidence that suggests myogenic responses (postauricular muscle artifact) might contribute to the musician-related differences observed in FFRs (Bidelman et al 2024, Frontiers in Neuroscience).

c. (lines 62-63) While years of music training relationships are more suggestive of a training effect rather than mere categorical group differences, it does not rule out inherent, genetic predispositions that could have contributed to differences in neural responses (e.g., Mankel & Bidelman, 2018, PNAS), speech processing behaviors (e.g., Swaminathan & Schellenberg 2017, Psychonomic Bulletin & Review), voice emotion recognition (Correia et al 2022 Emotion), the desire to pursue musical activities in the first place (e.g., Wesseldijk et al 2021, Psychological Science), and/or to stick with the training for longer than less-musically-inclined peers (see also review by Schellenberg & Lima 2024, Annual Review of Psychology).

i. For further discussions on genetic contributions, see: Wesseldijk et al 2023, Translational Psychiatry; Tan et al 2014, Frontiers in Psychology; Mosing et al 2014, Psychological Science; Hambrick & Tucker-Drob 2015, Psychonomic Bulletin & Review; Wesseldijk et al 2019, Developmental Psychology; etc.

Thank you for your suggestions. We have expanded our introduction to address each of your suggestions above. We have used track changes to highlight the new additions to the literature review. A copy of the added text has also been copied below:

Musical training appears to be associated with plasticity in both cortical and subcortical brain regions that process pitch, duration, and onset time of voice stimuli.For review see 39 For example, when presented with linguistic pitch patterns musicians have enhanced and more accurate frequency encoding in the inferior colliculus.40 It has also been shown that musicians have modulated inter-regional neural communication compared to non-musicians.41, 42 For example, in music/speech categorical processing musicians had increased activation in early primary auditory cortex whereas less experienced non-musicians had increased activation in downstream, higher-order linguistic brain areas such as the inferior frontal gyrus.41 These findings indicate that cortical and subcortical processes may be under different learning timescales or constraints whether in terms of short-term experiences or for long-term experiences such as musical training. Recent evidence suggests that the enhanced frequency-following response (a scalp-recorded neuroelectric brain recording that serves as a neural index of sound encoding in EEG) may be impacted by myogenic responses, such as a postauricular muscle artifact which may contribute to the musician related differences observed in frequency-following responses.43

Cortical and subcortical responses have been correlated to years of musical training39, 40, 44 establishing a relationship with length of musical training. A relationship between musical training and speech processing at a cortical level has also been demonstrated, where enhanced and earlier cortical responses to syllabic duration and voice onset time has also been measured in children.45, 46 Finally, years of musical training is positively correlated to faster learning of voice identification in a non-native language.47 English and Mandarin speaking musicians have an advantage for voice identity recognition for unfamiliar languages that may be attributed to superior pitch processing abilities.48 While years of music training relationships are suggestive of a training effect, recent studies have demonstrated that inherent, genetic predispositions may also contribute to differences in neural responses,49 speech processing behaviours,50 voice emotion recognition,51 the desire to pursue musical activities,52 and to commit to training for longer than less musically inclined peers.53 for review

Methods

4. Certain aspects of the methods are missing or not well described, making it difficult to evaluate the rigor and potentially limiting reproducibility of the study:

a. Even if the stimuli were controlled for RMS amplitudes, background noise, distinguishing features, etc. as mentioned in the methods, characteristic differences across the stimuli could still artificially inflate identification performance. Were the objects, faces, and associated audio validated for equivalent identification performance? Did all the voices speak the same English phrase? Do the car horns all have unique timbres and pitch contours, or are the pitch contours similar?

For this task objects, faces, and associated auditory stimuli were not separately validated for equivalent identification performance. Individual variability was reduced across all stimuli, both learned and new, to control for external factors such as background noise and distinguishing features

---

## [Decision Letter · Decision Letter 1]

9 Mar 2025

PONE-D-24-32230R1Enhanced voice recognition in musiciansPLOS ONE

Dear Dr. Moro,

Thank you for submitting your manuscript to PLOS ONE. After careful consideration, we feel minor revisions are required to meet PLOS ONE’s publication criteria. Therefore, we invite you to submit a revised version of the manuscript that addresses the points raised during the review process.

We look forward to receiving your revised manuscript.

Kind regards,

Gavin M. Bidelman, Ph.D.

Academic Editor

PLOS ONE

**Journal Requirements:**

Reviewers' comments:

Reviewer's Responses to Questions

**Comments to the Author**

1. If the authors have adequately addressed your comments raised in a previous round of review and you feel that this manuscript is now acceptable for publication, you may indicate that here to bypass the “Comments to the Author” section, enter your conflict of interest statement in the “Confidential to Editor” section, and submit your "Accept" recommendation.

Reviewer #1: (No Response)

Reviewer #2: All comments have been addressed

2. Is the manuscript technically sound, and do the data support the conclusions?

Reviewer #1: Yes

Reviewer #2: Yes

3. Has the statistical analysis been performed appropriately and rigorously? 

Reviewer #1: Yes

Reviewer #2: Yes

4. Have the authors made all data underlying the findings in their manuscript fully available?

Reviewer #1: Yes

Reviewer #2: Yes

5. Is the manuscript presented in an intelligible fashion and written in standard English?

Reviewer #1: Yes

Reviewer #2: Yes

6. Review Comments to the Author

**Reviewer #1:**  This manuscript revision shows significant improvements. Specifically, I noticed better clarity in voice/person recognition terminology (cf. speech or linguistic processing), the methods have been updated with additional details, and a new analysis was added to evaluate possible differences in stimulus category sensitivity across the participant groups. I have a few more comments below that, if addressed, would further clarify aspects of this manuscript.

Major

1. Lines 272-273 state: “There is no relationship between hours of weekly practice and

sensitivity to voices for person identification.” However, the stats in Table 1 and discussion suggest otherwise: e.g., lines 304-306: “this ability is related to the extent of experience as measured by hours of weekly practice and years of training”; lines 312-314: “better unimodal auditory sensitivity for voices is positively correlated with years of musical experience, as well as hours of practice”. Please resolve these contradictions.

2. Lines 294-295: “Non-musicians demonstrated a strong trend for enhanced auditory sensitivity for voices (M = 2.05, SD = 0.61, 95% CI [1.84, 2.26]) compared to object sounds”, but Figure 3 and lines 322-323 state “On the other hand, non-musicians show a strong trend for better sensitivity to object sounds compared to voices.” Please correct the first statement if the latter are indeed correct.

Minor

3. Line 73: Speech-in-noise cognition—is this intended to say “recognition”?

4. I am a little confused by the wording “by replacing their perfect score with a constant” (line 191). This suggests to me that scores of 100% were replaced by a value of 0.025, but I am not certain that was the intended meaning.

5. Line 198: “vsm” in parentheses—per the tracked changes version, I think this was meant to state “Musicians vs. Non-Musicians”.

6. The authors’ response letter suggests Figure 1 was kept in the manuscript, but I did not see the original figures 1 & 2 in my copy, nor the corresponding object recognition figure. In the first manuscript version, I did find these helpful as a reader. It would be nice to add them back in if they can still be included per editor/publishing requirements.

**Reviewer #2: ** The authors have satisfactorily addressed my previous comments. The article is a valuable contribution.

7. PLOS authors have the option to publish the peer review history of their article (what does this mean? ). If published, this will include your full peer review and any attached files.

**Do you want your identity to be public for this peer review?** For information about this choice, including consent withdrawal, please see our Privacy Policy .

Reviewer #1: No

Reviewer #2: No

---

## [Author Response · Author response to Decision Letter 2]

6 Apr 2025

Journal Requirements:

* Data are published on York University’s Dataverse. It can be found at the following DOI: https://doi.org/10.5683/SP3/L8OBJW

Response: Reference list has been reviewed and verified.

Reviewer #1: This manuscript revision shows significant improvements. Specifically, I noticed better clarity in voice/person recognition terminology (cf. speech or linguistic processing), the methods have been updated with additional details, and a new analysis was added to evaluate possible differences in stimulus category sensitivity across the participant groups. I have a few more comments below that, if addressed, would further clarify aspects of this manuscript.

Major:

1. Lines 272-273 state: “There is no relationship between hours of weekly practice and

sensitivity to voices for person identification.” However, the stats in Table 1 and discussion suggest otherwise: e.g., lines 304-306: “this ability is related to the extent of experience as measured by hours of weekly practice and years of training”; lines 312-314: “better unimodal auditory sensitivity for voices is positively correlated with years of musical experience, as well as hours of practice”. Please resolve these contradictions.

Response: Thank you for noticing this error. The results have been corrected to reflect the table and discussion.

2. Lines 294-295: “Non-musicians demonstrated a strong trend for enhanced auditory sensitivity for voices (M = 2.05, SD = 0.61, 95% CI [1.84, 2.26]) compared to object sounds”, but Figure 3 and lines 322-323 state “On the other hand, non-musicians show a strong trend for better sensitivity to object sounds compared to voices.” Please correct the first statement if the latter are indeed correct.

Response: You are correct that the latter statements are correct. The results have been corrected to reflect this.

Minor:

3. Line 73: Speech-in-noise cognition—is this intended to say “recognition”?

Response: Corrected.

4. I am a little confused by the wording “by replacing their perfect score with a constant” (line 191). This suggests to me that scores of 100% were replaced by a value of 0.025, but I am not certain that was the intended meaning.

Response: Yes, that is correct. Perfect scores were adjusted by either adding 0.025 (in the case where the participant got 0) or subtracting 0.025 (in the case where the participant got 100). We have attempted to clarify this by updating the text to the following: Participants with perfect scores were adjusted based on recommendations by Macmillan and Creelman [60], by adjusting their score by a constant value (-0.025 in the case where participants scored 100).

5. Line 198: “vsm” in parentheses—per the tracked changes version, I think this was meant to state “Musicians vs. Non-Musicians”.

Response: Corrected.

6. The authors’ response letter suggests Figure 1 was kept in the manuscript, but I did not see the original figures 1 & 2 in my copy, nor the corresponding object recognition figure. In the first manuscript version, I did find these helpful as a reader. It would be nice to add them back in if they can still be included per editor/publishing requirements.

Response: In this case there is a conflict with editor/publishing requirements. Due to copyright issues for the car stimuli images we have had to remove the figures from the paper. We have addressed your comment while still adhering to publishing requirements by re-inserting the schematic diagrams but with different car images from an open source dataset to be used for illustrative purposes.

Reviewer #2: The authors have satisfactorily addressed my previous comments. The article is a valuable contribution.

Response: Thank you!

---

## [Editor Report · Decision Letter 2]

11 Apr 2025

Enhanced voice recognition in musicians

PONE-D-24-32230R2

Dear Dr. Moro,

We’re pleased to inform you that your manuscript has been judged scientifically suitable for publication and will be formally accepted for publication once it meets all outstanding technical requirements.

Kind regards,

Gavin M. Bidelman, Ph.D.

Academic Editor

PLOS ONE
---

## [Editor Report · Acceptance letter]

PONE-D-24-32230R2

PLOS ONE

Dear Dr. Moro,

I'm pleased to inform you that your manuscript has been deemed suitable for publication in PLOS ONE. Congratulations! Your manuscript is now being handed over to our production team.

Kind regards,

on behalf of

Dr. Gavin M. Bidelman

Academic Editor

PLOS ONE